# Resting Energy Expenditure and Related Factors in 6- to 9-Year-Old Southern African Children of Diverse Population Groups

**DOI:** 10.3390/nu13061983

**Published:** 2021-06-09

**Authors:** Adeline Pretorius, Paola Wood, Piet Becker, Friedeburg Wenhold

**Affiliations:** 1Department Human Nutrition, Faculty of Health Sciences, University of Pretoria, Private Bag X323, Arcadia 0007, South Africa; 2Department of Physiology, Faculty of Health Sciences, University of Pretoria, Private Bag X323, Arcadia 0007, South Africa; paola.wood@up.ac.za; 3Research Office, Faculty of Health Sciences, University of Pretoria, Private Bag X323, Arcadia 0007, South Africa; piet.becker@up.ac.za

**Keywords:** resting energy expenditure, indirect calorimetry, body composition, physical activity, population group, obesity

## Abstract

Lower resting energy expenditure (REE) may partially explain the disproportionate prevalence of overweight/obesity among black African women. As no previous studies have investigated the REE of Southern African (South. Afr.) children, we aimed to determine, by sex and population group, the REE of 6- to 9-year-old urban school children. In a cross-sectional study with quota sampling, REE was measured with indirect calorimetry (IC). Confounders considered were: body composition (BC) (fat-free mass (FFM), FFM index, fat mass (FM), FM index), assessed using multifrequency bioelectrical impedance analysis, and physical activity (PA) measured with a pedometer. Multivariate regression was used to calculate REE adjusted for phenotypes (BC, z-scores of weight-for-age, height-for-age, body mass index-for-age) and PA. Sex and population differences in REE were determined with two-way ANOVA. Ninety-four healthy children (59.6% girls; 52.1% black) with similar socioeconomic status and PA opportunities participated. Despite BC variations, sex differences in REE were not significant (41 kcal/day; *P* = 0.375). The REE of black participants was lower than of white (146 kcal/day; *P* = 0.002). When adjusted for FFM and HFA z-score, the differences in REE declined but remained clinically meaningful at 91 kcal/day (*P* = 0.039) and 82 kcal/day (*P* = 0.108), respectively. We recommend the development of population-specific REE prediction equations for South. Afr. children.

## 1. Introduction

Overnutrition is a rising epidemic worldwide and children are increasingly affected [1]. For young children, being overweight or obese can have direct physical and mental health effects and is considered a major risk factor for diet-related non-communicable diseases (NCDs) during adulthood [2]. Although overweight and obesity affect all populations, some population groups, e.g., black Africans and African Americans, may be disproportionally affected [3,4]. In South Africa, there is a great need to better understand overweight, obesity and NCDs and the related risk factors, and to transform this information into effective healthcare policies, programmes and services [5]. Since children are the future generation, population-specific and timely nutrition and physical activity (PA) intervention will contribute to the future health of all population groups [2]. Overweight and obesity are the result of a positive energy balance when energy intake exceeds total energy expenditure for an extended period of time. Resting energy expenditure (REE) is a major and, for most individuals, the largest component of total energy expenditure [6]. Indirect open-circuit calorimetry (IC) is the most practical and widely employed method used to measure REE [7]. Many studies [8,9,10,11,12,13] suggest that lower REE may contribute to the higher prevalence of overweight and obesity in black African/African American individuals. Additionally, the degree of African versus European ancestry may have an amplified effect on REE, meaning that black African groups from Sub-Saharan countries may have an even lower REE than African American groups, thereby increasing their risk of overweight and obesity [14,15]. Research to investigate the REE in Sub-Saharan population groups is limited [14]. In South Africa, two recent studies found lower REE in black adults compared to their white counterparts [16,17]. Studies to determine REE differences between children of various population groups are also limited to the African American and white populations and have indicated varied results [8,9,11,12,18,19,20]. No known studies have investigated the REE of Sub-Saharan children. Such research could provide an indication of whether population differences in REE are “acquired” later in life (during adulthood) or whether they already exist at a relatively young age.

When assessing the REE of children, factors that may influence REE should be considered to ensure appropriate interpretation. Age, sex, body size and body composition, specifically fat-free mass (FFM), have been identified as the most significant factors, with population group and PA as possible contributing factors [8,9,21,22,23]. After REE, PA is the second largest and most variable component contributing to total energy expenditure. It refers to any bodily movements, both voluntary and involuntary, produced by muscle contraction [6]. Pedometers are generally used as an objective and affordable tool to measure PA of children by registering the number of steps during walking and running [24]. Apart from the direct effect of PA on total energy expenditure, evidence exists [25,26] that PA may influence REE, and its effects can last for hours or days (referred to as the excess post-exercise O_2_ consumption [EPOC]) [6,26,27]. It has been reported [25] that a lower REE in African American than white women may partially be explained by their lower fitness associated with a lower PA level. However, to our knowledge, no such studies have been performed in children.

Nested in an umbrella study of REE and physical activity (PA) of a diverse group of 6- to 9-year-old children, this study aimed to determine, by sex and population group, the REE of black and white Southern African (South. Afr.) children attending two primary schools in the City of Tshwane metropolitan area, Gauteng province in South Africa, taking phenotypic characteristics (body size and body composition) and PA as confounders and the children’s socioeconomic context into account.

## 2. Materials and Methods

### 2.1. Study Design, Population and Sampling

In this cross-sectional study, the recruitment of participants aimed to include equal numbers of 6- to 9-year-old prepubertal black and white South. Afr. boys and girls attending two primary schools. Although socioeconomic status (SES) is not generally recognised as a determinant of REE, some evidence suggests [13,28] an adaptive response in REE to nutritional constraints. The two schools were therefore purposively selected based on their attendance of mainly black and white children, their perceived similarity (before the study) in the SES of the schools and of families who generally enrol their children at these schools, and in the access to PA opportunities at the school. Self-identification of population is considered a good approximation of ancestral origin [29], and was used for classification of population group. Nationalities from South. Afr. countries, i.e., all countries south of the equator, were included. Children were considered eligible based on parental report of the country of birth of both parents and the nationality of the child. Age in years to the nearest month was calculated based on the reported date of birth and date of assessment. 

Participants were invited via a letter distributed to parents of all learners aged 6- to 9-years at the two schools. Two venues, one close to the school and the other on the school grounds were used. All measurements were taken early in the morning (06 h30–07 h30) before school. Recruitment continued until the required sample size was reached. Comparison of population groups with respect to REE, overall and within sex categories, was considered for sample size calculations. To detect a clinically relevant difference in REE of 75 kcal/day between groups and assuming a common SD of 84 kcal/day [16,17], a sample of 28 participants per group was calculated to have 90% power when using a two-sided two-group Student’s paired t-test at the 0.05 level of significance. More accurate calculation of sample size before the onset of the study, and the estimation of oversampling requirements to overcome sampling challenges and non-compliance, were not possible since REE research in South. Afr. children was not available. Consequently, the power of the sample was recalculated after data collection to ensure the power of the sample remained at least 90%. Since two subgroup analyses were of interest, i.e., for boys and for girls, 60 participants from each population group were enrolled into the study for a total sample size of 120 participants (69 girls of which 32 were black; 28 black boys).

### 2.2. Data Collection and Management

A descriptive audit of the two schools’ built environment and surrounding neighbourhoods was performed using the international study of childhood obesity, lifestyle and environment [30] school audit tool, ISAT [31]. The ISAT assesses aspects including buildings, amenities and areas linked to PA. Since children spend a great amount of time at school, the school environment is considered a significant factor that may influence health-related behaviour, including PA, of young children [32,33,34]. Therefore, the school’s built environment is often used to describe access to PA facilities and opportunities that may enhance the PA of children [34]. 

Before a child’s appointment, consenting parents were sent an information letter with prerequisites for REE measurements [26,35,36,37], including the absence of injuries, illness and use of chronic medication, a minimum of 7 h sleep the night before, 12 h of abstinence from PA and no intake of food or beverages (except water) since 9 pm the night before. Apart from the informed consent parents completed a sociodemographic questionnaire to identify the familial SES based on income category, housing conditions, household services (water, sanitation, electricity supply), employment status and level of education. 

On arrival, the study purpose and measuring procedures were individually explained and demonstrated to each participant and assent was obtained. To help put the participants at ease, familiar measurements were taken first including height, followed by body weight and bioelectrical impedance analysis (BIA) to assess body composition. The weight and height were measured barefoot in a light weighted school uniform, following the protocol of the United States Centers for Disease Control and Prevention [38]. The Seca 274 digital mobile stadiometer (Hamburg, Germany) [39] was used to measure standing height (cm) to the nearest 0.1 cm. The reading was transferred wirelessly to the multifrequency Seca mBCA 514 (Hamburg, Germany) [39], which was used to measure weight (kg), to the nearest 100 g, and BIA. Height, weight and BIA data were transferred wirelessly from the Seca mBCA to a personal computer. Z-scores of weight-for-age (WFA), height-for-age (HFA) and body mass index for age (BMI-FA) were calculated by the built-in software from Seca mBCA 514 based on the WHO growth reference data [40] and displayed in Microsoft Excel. The resistance (R) value of the BIA, measured at 50 Hz, was used to calculate body composition. The resistance values were manually transferred to Microsoft Excel to calculate FFM (kg) by using the equation of Horlick [41]: 

FFM (kg) = [(0.459 × H^2^/R) + (0.064 × W) + 3.474]/[(0.769 − (0.009 × age) − (0.016 × sex)] (H = body height in cm; R = resistance in Ω; W = body weight in kg).

Fat mass [3] was calculated by subtracting the calculated FFM from total body weight (kg). Fat-free mass index (FFMI) and fat mass index (FMI) were, respectively, calculated by dividing FFM and FM (both in kg) by height in metres squared. 

Resting energy expenditure (kcal/day) was measured using the Cosmed Quark RMR indirect calorimeter (Rome) [42] with a ventilated canopy hood and disposable antibacterial filter. A full service of all measurement instruments prior to the study, and daily calibration were performed according to manufacturer instructions [39,42].

Achieving and maintaining steady state during IC is recommended to improve measurement validity [7,35,43]. Steady state was monitored using the Quark RMR machine-generated indicator, defined by the manufacturer [42] as a period when the average minute VO_2_ and VCO_2_ changes by less than 10% and the average RQ changes by less than 5%. Evidence-based measurement protocols are advised to ensure an individual reaches a complete resting state and maintains the required steady state during the REE measurement [35]. Due to the lack of a standardised REE measurement protocol for children [35,36,44], an abbreviated adult protocol, along with recommendations based on previous REE measurements of children [35,36,44], were used for this study: Measurements were taken without a prior rest period, in contrast to recommendation for adults, and continued for 15–20 min depending on the participant’s resting state. Participants were measured in a supine position, hands by their side and legs extended forward. An age-appropriate audible short story was played to keep participants still and relaxed throughout the measurement while being observed by the principal investigator. Room temperature was maintained between 20–25 °C in a quiet venue with dimmed lighting. All values prior to the achievement of machine-indicated steady state were discarded before the average REE values per participant were calculated. 

A spring-levered Yamax Digi-Walker SW-800 pedometer (Yamasa, Japan) was used to measure PA. Following IC measurements, the pedometer was securely fitted around the child’s waist to prevent tilting, thereby improving accuracy [45]. The device was covered with masking tape to prevent reactivity and device tampering [45,46,47,48]. Wearing instructions were provided, including securely fitting the pedometer at the time of rising in the morning until going to bed at night, correct positioning and daily covering with masking tape. Parents were requested to inform the principal investigator via a text message, of the step count reading at the end of each day, and participants were randomly monitored after school to ensure wearing instructions were adhered to. The pedometer was collected from participants after seven consecutive days of wear. For each participant, the average number of steps/day was calculated.

### 2.3. Data Analysis

Data were transferred from Microsoft Excel to Stata Statistical Software Release 15 (StataCorp, 2017, StataCorp LCC, College Station, TX, USA) for statistical analysis. The significance level was set at 0.05 for all analyses. Two-sided Fisher’s exact test was used to assess and describe differences in SES categorical variables between population groups and between schools. Continuous variables (age, phenotypic variables, REE and PA) were summarised by sex and population group, reporting linear estimated means (predictive margins) including a 95% confidence interval, following an analysis of variance (ANOVA) with fixed effects sex, population group and their interaction. The two-way ANOVA was further used to examine the effect of sex and population group and their interaction on the measured REE. Spearman’s product-moment correlation, due to non-normal distribution of variables, was used to determine the strength and statistical significance of the relationship between REE and each relevant variable (age, WFA z-score, HFA z-score, BMI-FA z-score, FFM, FFMI, FM, FMI, average steps/day). With multivariable regression, the associations of sex and population group were assessed with respect to REE, taking interaction into account of these same covariates. 

## 3. Results

### 3.1. Description of the Sample

The data of 26 participants were excluded from the initial 120 recruited, resulting in a final sample size of 94 (Table 1). The data of 19 children were excluded since steady state was not achieved during the REE measurement, and seven participants were excluded due to implausible data, e.g., power failure during a measurement or outlier REE values (below 420 kcal/day). 

The age of participants ranged from 6.4 to 9.7 years, with a mean of 7.9 years (SD = 0.79). The results of the ISAT confirmed that the built environment of both schools was similar in terms of opportunities to PA. Both schools and their surrounding neighbourhoods supported motorised transport but provided suboptimal facilities to support walking or cycling to school. Both offered multiple access to good quality sports and recreational amenities to encourage informal play as well as organised sports activities. Several playgrounds, sports areas and opportunities to interact with nature were also available at both schools, with an abundance of trees and gardens. School grounds appeared neat and clean and provided a pleasing aesthetic environment. Most households (90.0%) were within the upper quintile of income, with two to three people per household earning an income and no unemployment. All caregivers had completed high school, and 95.7% completed tertiary education. Housing conditions (home ownership and number of members per household) and access to household services reflected a higher living standard (Table 2). The SES of the two schools was considered homogenous, since the differences in income categories, home ownership and number of household members, access to household services, and level of education between schools and between population groups did not meet statistical significance for any of the categories of SES (all *P* > 0.05).

Table 3 summarises the phenotypic characteristics and the average steps/day of the sample. Between the sexes, significant differences were observed in body composition variables, with the mean FFM and FFMI significantly (*P* < 0.001 for both) lower in girls than in boys. Conversely, the mean FM and FMI were significantly higher (*P* = 0.035 and *P* = 0.010, respectively) in girls than boys. Between population groups, the mean HFA z-score and FFM were significantly lower (*P* < 0.001 and *P* = 0.003, respectively) for the black children. In contrast, the BMI-FA z-score, FM and FMI were higher for the black group. The mean average steps/day for black participants was significantly lower (*P* < 0.001) than for white participants, and similarly, the mean for girls was lower than for boys (*P* = 0.029). 

Based on the WHO growth reference data [40], the majority of the participating children (83.0%) had a healthy weight (−2 ≤ BMI-FA z-score ≤ 1). A higher percentage of black than white children were overweight (1 < BMI-FA z-score ≤ 2), 14.3% and 4.5%, respectively, or obese (BMI-FA z-score > 2), 12.2% and 2.2%, respectively. More girls (12.5%) than boys (5.3%) were overweight, while the obese children were almost equally distributed between girls (7.1%) and boys (7.9%). All the white boys in the sample had a healthy weight.

### 3.2. Factors Related to Measured REE

Table 4 summarises the relationship between mean measured REE (reported in Table 5) and age, phenotypic factors and PA, respectively. 

For the total sample, a significant relationship (*P* < 0.05) existed between REE and all phenotypic variables, except for FM and FMI, where a positive and significant relationship existed for black participants only. In addition, for black participants in particular, a positive, significant relationship existed between REE and anthropometric measurements including WFA z-score (r = 0.56; *P* < 0.001), HFA z-score (r = 0.32; *P* = 0.026) and BMI-FA z-score (r = 0.55; *P* < 0.001), respectively. The correlation between REE and FFM was the strongest (r = 0.45) and significant for all subgroups (*P* < 0.05), but for boys and for white participants the significance declined (*P* > 0.05) when the relationship between REE and FFMI was determined. The relationship between measured REE and PA lacked significance across subgroups.

### 3.3. Measured and Adjusted REE

There was no significant interaction between the effects of sex and population group on REE (F(1, 90) = 0.54; *P* = 0.465). No statistically significant difference in REE between sex categories was observed (F(1, 90) = 0.74; *P* = 0.393), but statistical significant differences existed between population groups (F(1, 90) = 9.34; *P* < 0.005).

Table 5 shows and confirms that the difference in REE, measured and adjusted for all phenotypic variables and PA, between boys and girls did not meet statistical significance. Between population groups, the REE remained significantly lower (*P* < 0.05) for black than white participants, even after REE was adjusted for phenotypic variables as well as PA, except when REE was adjusted for HFA z-score (*P* = 0.108).

## 4. Discussion

Our results indicate that the mean measured REE of 6- to 9-year-old black children was lower than for their white counterparts. However, for more in-depth interpretation, factors known to potentially act as confounders were considered. These are discussed below in relation to existing literature.

### 4.1. Age

We found that within the narrow age range we studied, age was not related to the measured REE of the total sample. When adjusted for age, the mean REE across sex and population subgroups, remained unchanged. This is in line with existing evidence [21,49] and in particular holds true for children in middle childhood across sex categories and the population groups we investigated.

### 4.2. Sex

Several studies [21,22,23,50,51] recognised sex as a significant predictor of REE in children of all ages, with boys having a higher measured REE than girls. These differences, often to a lesser extent, persisted after adjustment for influencing factors, including body size and body composition. Maffeis et al. [23], in their study of 6- to 10-year-old children, reported a mean measured REE of 40 kcal/day lower for girls than boys, with differences in FFM explaining 64% of the variability in REE. Further evidence exists [11,52] that when adjusted for body weight or FFM, the effect of sex on REE is negligible among prepubertal children younger than 12 years. Similar to these reports, our study indicated that the mean measured REE for girls was 41 kcal/day lower than for boys and did not meet statistical significance. When adjusted for WFA z-score, HFA z-score or BMI-FA z-score, REE values remained almost unchanged for both boys and girls, and the difference between sexes still did not meet statistical significance. When adjusted for FFM and FFMI, respectively the difference in REE was not only lower but changed direction with girls having a higher adjusted REE than boys (25 and 31 kcal/day; *P* = 0.582 and 0.549, respectively). These results are consistent with the findings of Lazzer et al. [22], who reported that a higher basal metabolic rate (BMR) (BMR can be extrapolated to REE) in boys could be explained mainly by their higher FFM when compared to girls. They further suggested that the confounding effects of sex may be eliminated when BMR is adjusted for FFM. The sex differences (although not statistically significant) in measured REE observed in our study may therefore be partially explained by the higher FFM in boys.

When adjusted for FM and FMI, respectively the REE difference between sexes increased. This conforms with previous research [11,52], indicating that FM also contributes to REE. It appears that in our study, the higher FM and FMI index of girls compared to boys contributed to their REE, thereby decreasing the sex difference in the measured REE.

Notwithstanding the above, the measured and adjusted sex differences in REE did not meet statistical significance. The small differences observed in the sample can be explained by differences in body composition.

### 4.3. Phenotype

The sample represented a healthy population in terms of physical growth, with most of participants (83.0%) falling under the WHO healthy weight category [1]. The mean WFA z-scores were relatively consistent across sex categories and population groups. Black participants had a significantly lower HFA z-score (*P* < 0.001) than the white group. Along with overweight and obesity, underweight and stunting have been identified as the most common nutritional disorders in young children, especially in black African populations living in less affluent rural settings [4,53]. However, in our study no evidence of stunting was observed, since the minimum HFA z-score of −1.54 SD was within the healthy range (greater than −2 SD) [40]. This could be explained by the high SES and the urban setting of this study, and the dramatic decrease in the prevalence of stunting reported during the past decades, especially for children above the age of 5 years living in urban areas [4]. When considering BMI-FA z-scores, the mean for black participants were significantly higher than for white participants (*P* = 0.042). Although slightly above the median of the health reference values (0.43 SD), it was still within the healthy BMI range (−2 SD ≤ BMI-FA ≤ 1 SD) [40]. There were no significant differences in anthropometric indices between the girls and boys of this sample.

In line with previous research [11,21,50] the strongest, yet moderate, correlation in the total sample was observed between REE and FFM. Although the significance of this correlation persisted across sex categories and population groups, a stronger correlation was observed for boys and for the white population group. However, the strength of the correlation was considerably lower for boys and practically absent for white participants when FFM was expressed per height unit as FFMI, whereas the correlation remained unchanged for girls and was slightly stronger for black participants. This reveals the possibility that the higher REE of the white population group, and in particular boys, may be related to a higher FFM associated with their taller stature, as indicated by their higher HFA z-score.

When considering the correlation between REE and FM, a significant and positive relationship existed only for black participants, and the relationship remained almost unchanged when FM was expressed per unit of height, i.e., FMI. This may indicate that in black participants, FM contributed to REE, irrespective of height differences.

For black participants only, the strength of the relationship observed between REE and FM and FMI, respectively was very similar to the relationship between REE and FFM and FFMI, respectively. Both FM and FFM, independent of height, may therefore be equally and positively related to the REE of black children. Similar to the black participants, girls had a significantly higher FM and FMI than boys. However, contrary to expectation, the correlation between REE and FM as well as FMI, lacked significance for girls. In addition, for black participants, the relationship between REE and WFA z-score and BMI-FA z-score, respectively, was even stronger than that observed for FM and FFM, but this was not mirrored in the white participants. It therefore appears that WFA and BMI (reflective of total body weight), and FM and FFM may contribute more to REE for black participants than is the case of white participants. This is not completely surprising, since individual variances in metabolic activity of FM and FFM in adults have been reported in the literature [54]. Metabolic activity of FFM is not consistent across all organs and skeletal muscle [13,55,56], and in children the effect of growth on the size of FFM and its composition may further contribute to variations in REE [13,57,58].

### 4.4. Population Group

The differences observed in the measured REE between population groups (146 kcal/day) decreased by a mere 18 kcal/day after adjustments were made for WFA z-score as well as for FFMI, suggesting that weight and FFMI did not contribute in practical terms to the differences in REE between population groups. When adjusted for FFM and HFA z-score, respectively, the population differences were considerably larger (91 kcal/day; *P* = 0.039 and 82 kcal/day; *P* = 0.108, respectively lower for the black children) and confirms that variances in FFM and height can partially explain differences in REE between population groups. Although the difference of 82 kcal/day did not meet statistical significance, it may have clinical implications. It has been proposed that as little as 50–100 kcal/day could affect long-term body weight and related health consequences in adults [59]. If the same principle applies to children, an excess of around 80 kcal/day (7.8% of mean measured REE) may lead to long-term health consequences.

Adjustments for BMI-FA z-score, FM and FMI increased the population difference (*P* < 0.001). This may indicate that phenotypic differences may have somewhat contributed to the observed population differences. These differences remained significant after being adjusted, indicating that the mean REE of black participants was lower, despite their higher BMI-FA z-score, FM and FMI.

The study results and the preceding discussion indicate that the REE of black participants was lower (statistically or clinically) than for white participants, even after being adjusted for all phenotypic variables. These results are in line with previous studies [8,11,18,19] performed on mainly American children. However, it is often argued that the lower REE of African American population groups may be explained by differences in the type and distribution of FFM rather than an absolute difference in REE [18,58,60]. Since FFM is considered the most metabolically active tissue, variations in metabolic activity between types of muscle mass may lead to considerable differences in REE [55]. Broadney et al. [18] indicated that the REE of black African children aged 5–18 years was 77 ± 16 kcal/day lower than their white counterparts after being adjusted for lean mass (skeletal muscle and organ mass). When adjusted for trunk and appendicular lean mass, the difference declined to 28 ± 19 kcal/day. Similar findings were observed by Tershakovec et al. [51], reporting that the difference in REE between African American and white children declined from 111 kcal/day to 77 kcal/day when adjusted for FFM and trunk FFM, respectively. Many studies [18,51,55,58,60] indicate that African American populations have less of the metabolically active trunk lean mass, consistent with lower organ mass, which may explain a lower REE than in white populations. In addition, a greater appendicular lean mass, which contributes little to energy expenditure at rest, along with a higher metabolically inactive bone mass in African American groups, may further reduce REE [19,61]. It may therefore be possible that a lower trunk lean mass and higher bone mass of the black participants in our study may have contributed to the differences in REE. However, these arguments were contradicted by the research of Adzika et al. [14], who reported that for adults, the REE of Sub-Saharan black African populations was 240–275 kcal/day lower than the REE of white groups, even after adjustments were made for trunk and appendicular lean mass and for bone mass. It was consequently suggested that a lower REE among black African participants may be related to genotype, since it had previously been identified [15] that European ancestry may be related to a higher REE. No known studies have been undertaken to explore the effect of BC compartments or genotype on REE in South. Afr. children and further investigation may be required to explore the reasons for population differences in REE.

Regardless of factors contributing to REE, acknowledging differences in REE between population groups is essential for acceptable estimation of REE in the prevention and management of childhood obesity. Previous studies at our institution [16,17] identified a lower REE in the black than white population group, specifically for women, which may contribute to the high prevalence of overweight or obesity in black South African women [4]. Our study results suggest that sex-associated population differences in REE may only occur after middle childhood but prior to adulthood and constitutes a future research question.

### 4.5. Physical Activity

Physical activity can affect REE in various ways. It is well known that an increase in PA affects metabolic processes, thereby increasing BMR and REE for up to 48 h after an exercise session (EPOC) [26,35]. It is for this reason that abstinence from PA before REE measurements is required [35]. Additionally, regular PA may result in muscle growth. Since FFM is related to REE, regular PA may indirectly increase REE [6]. Apart from these influences of PA on REE, previous yet limited research [62] indicated the possibility of a long-lasting effect of PA on REE beyond the effect of EPOC.

Although no gold standard device has been identified to objectively measure PA, accelerometers are often the preferred option. However, these devices may still include measurement errors and are relatively costly [63,64]. Pedometers, an inexpensive alternative for the objective measurement of PA, along with a detailed measurement protocol as applied in our study, are considered reliable and accurate when compared to accelerometers [45].

In the present study, no significant correlation was observed between REE and PA, and the population difference in REE remained significant, even when it was adjusted for daily step-count. It can therefore be concluded that the population difference in REE of this sample was not related to differences in PA.

### 4.6. Strengths and Limitations

The study was the first known to investigate the REE of children in South. Afr. and represents two population groups, white (for whom most prediction equations are developed) as well as black (the majority population in the country with the highest prevalence of female overweight and obesity). This is the first known study to investigate the relationship between REE and PA of children.

Resource-limitations necessitated the use of BIA instead of a gold standard measure such as dual energy X-ray absorptiometry (DEXA) for body composition analysis, and pedometers were used instead of accelerometery for the measurement of PA. Additionally, parental self-report measures, instead of clinical assessments, were used to determine health and, thus, underlying conditions such as a common cold, attention deficit disorder or other illnesses of participants that may have affected REE.

## 5. Conclusions and Recommendations

Our study indicates that the REE of 6- to 9-year-old black children in South. Afr. is lower than for their white counterparts, regardless of phenotypic characteristics and PA. Between-sexes differences in REE were not significant. Using generally available, non-population-specific equations to estimate REE may result in ineffective prevention and treatment of overweight and obesity. In South Africa, actual measurement of REE is not feasible in most clinical and research settings. Population-specific prediction equations should be developed taking the factors identified in our study into consideration to more accurately estimate the energy requirements of South. Afr. children.

## Figures and Tables

**Table 1 nutrients-13-01983-t001:** Distribution of the sample by sex and population group (*N* = 94).

	Black	White	Total Sample
Girls: *n* (%)	27 (48)	29 (52)	56
Boys: *n* (%)	22 (58)	16 (42)	38
Total: *n* (%)	49 (52)	45 (48)	94

**Table 2 nutrients-13-01983-t002:** Sociodemographic distribution of the sample by population group (*N* = 93) ^a^.

Sociodemographic Variable		Black	White	Total Sample
Income category	Upper quintile: *n* (%)	39 (81)	45 (100)	84 (90)
	3rd quintile: *n* (%)	2 (4)	0	2 (2)
	Lower quintile: *n* (%)	2 (4)	0	2 (2)
	Did not indicate: *n* (%)	5 (11)	0	5 (6)
Level of education	High school: *n* (%)	3 (6)	0	3 (3)
	Tertiary education: *n* (%)	44 (92)	45 (100)	89 (96)
	Did not indicate: *n* (%)	1 (2)	0	1 (1)
Home ownership	Home owners: *n* (%)	35 (73)	35 (78)	70 (75)
	Renting: *n* (%)	11 (23)	10 (22)	21 (23)
	Live with family/friends: *n* (%)	2 (4)	0	2 (2)
Number of members per household	Two: *n* (%)	2 (4)	0	2 (2)
	Three–five: *n* (%)	39 (81)	38 (85)	77 (82)
	Six–eight: *n* (%)	7 (15)	6 (13)	13 (15)
	Nine or more: *n* (%)	0	1 (2)	1 (1)
Household services	Water, sanitation, electricity supply: *n* (%)	48 (100)	45 (100)	93 (100)

^a^ For one black participant the sociodemographic questionnaire was not completed.

**Table 3 nutrients-13-01983-t003:** Phenotypic description and PA of the sample by sex and population group (*N* = 94) ^a^.

	Sex	Mean ^b^	95% CI ^c^	Sex Difference (Boys–Girls)	*P*-Value ^d^	Population Group	Mean ^b^	95% CI ^c^	Population Difference (Black–White)	*P*-Value ^d^
WFA z-score	Girls	0.49	(0.23; 0.75)	0.00	0.997	Black	0.38	(0.10; 0.66)	−0.22	0.270
	Boys	0.49	(0.17; 0.81)			White	0.61	(0.32; 0.90)		
HFA z-score	Girls	0.53	(0.31; 0.75)	0.04	0.820	Black	0.08	(−0.16; 0.32)	−0.97	<0.001
	Boys	0.57	(0.30; 0.84)			White	1.05	(0.81; 1.30)		
BMI-FA z-score	Girls	0.23	(−0.05; 0.52)	−0.06	0.795	Black	0.43	(0.12; 0.73)	0.46	0.042
	Boys	0.17	(−0.17; 0.52)			White	−0.03	(−0.35; 0.29)		
FFM (kg)	Girls	20.14	(19.54; 20.74)	1.73	<0.001	Black	20.15	(19.50; 20.79)	−1.45	0.003
	Boys	21.87	(21.14;22.61)			White	21.60	(20.92; 22.27)		
FFMI (kg/m^2^)	Girls	12.01	(11.80; 12.22)	0.93	<0.001	Black	12.28	(12.06; 12.50)	−0.23	0.166
	Boys	12.94	(12.69; 13.20)			White	12.51	(12.27; 12.74)		
FM (kg)	Girls	7.49	(6.48; 8.50)	−1.71	0.035	Black	7.82	(6.74; 8.89)	2.12	0.008
	Boys	5.78	(4.55; 7.01)			White	5.70	(4.57; 6.82)		
FMI (kg/m^2^)	Girls	4.44	(3.92; 4.96)	−1.08	0.010	Black	4.66	(4.11; 5.22)	1.37	<0.001
	Boys	3.37	(2.73; 4.00)			White	3.29	(2.71; 3.87)		
Average steps/day	Girls	10,212	(9519; 10,906)	1220	0.029	Black	9280	(8538; 10,022)	−2979	<0.001
	Boys	11,433	(10,588; 12,277)			White	12,258	(11,483; 13,033)		

WFA Weight-for-age; HFA Height-for-age; BMI-FA Body mass index-for-age; FFM Fat-free mass; FFMI Fat-free mass index; FM Fat mass; FMI Fat mass index. ^a^ Girls *n* = 56; boys *n* = 38; black *n* = 49; white *n* = 45. ^b^ Adjusted mean: Predictive margins of the general linear model for ANOVA with the factors sex and population group and their interaction. ^c^ 95% Confidence interval around the mean. ^d^ Two-way ANOVA.

**Table 4 nutrients-13-01983-t004:** Relationship between mean measured REE and age, phenotypic factors and PA (*N* = 94) ^a^.

Variable	Sex	R ^b^	*P*-Value ^c^	Population Group	R ^b^	*P*-Value ^c^	Total Sample	
							R ^b^	*P*-Value ^c^
Age	Girls	−0.27	0.050	Black	0.34	0.021	−0.08	0.440
	Boys	0.16	0.350	White	−0.34	0.016		
WFA z-score	Girls	0.32	0.016	Black	0.56	<0.001	0.37	<0.001
	Boys	0.41	0.011	White	−0.02	0.881		
HFA z-score	Girls	0.37	0.006	Black	0.32	0.026	0.36	<0.001
	Boys	0.36	0.025	White	0.13	0.383		
BMI-FA z-score	Girls	0.15	0.281	Black	0.55	<0.001	0.21	0.045
	Boys	0.27	0.100	White	−0.10	0.520		
FFM (kg)	Girls	0.35	0.008	Black	0.39	0.006	0.45	<0.001
	Boys	0.55	<0.001	White	0.42	0.004		
FFMI (kg/m^2^)	Girls	0.35	0.008	Black	0.44	<0.001	0.30	0.003
	Boys	0.28	0.089	White	0.09	0.581		
FM (kg)	Girls	0.06	0.669	Black	0.39	0.005	0.17	0.105
	Boys	0.28	0.086	White	0.05	0.752		
FMI (kg/m^2^)	Girls	0.03	0.803	Black	0.41	0.004	0.13	0.227
	Boys	0.24	0.153	White	−0.04	0.772		
Average steps/day	Girls	0.09	0.488	Black	−0.09	0.558	0.05	0.651
	Boys	−0.01	0.958	White	−0.16	0.286		

WFA Weight-for-age; HFA Height-for-age; BMI-FA Body mass index-for-age; FFM Fat-free mass; FFMI Fat-free mass index; FM Fat mass; FMI Fat mass index. ^a^ Girls *n* = 56; boys *n* = 38; black *n* = 49; white *n* = 45. ^b^ Spearman’s product-moment correlation. ^c^ Level of statistical significance.

**Table 5 nutrients-13-01983-t005:** Mean REE (kcal/day) as measured and adjusted for covariates by sex and population groups (*N* = 94) ^a^.

REE-Related Variable	Sex	Mean ^b^	95% CI ^c^	Sex Difference (Boys–Girls)	*P*-Value ^d^	Population Group	Mean ^b^	95% CI ^c^	Population Difference (Black–White)	*P*-Value ^d^
Measured REE (kcal/day)	Girls	1005	(948; 1062)	41	0.375	Black	951	(890; 1013)	−146	0.002
	Boys	1045	(976; 1115)			White	1097	(1033; 1161)		
**REE adjusted ^e^ for:**										
Age	Girls	1005	(947; 1063)	41	0.377	Black	951	(889; 1014)	−146	0.002
	Boys	1045	(975; 1116)			White	1097	(1032; 1163)		
WFA z-score	Girls	1005	(951; 1059)	41	0.347	Black	960	(902; 1018)	−128	0.003
	Boys	1046	(980; 1111)			White	1088	(1028; 1149)		
HFA z-score	Girls	1006	(950; 1061)	38	0.393	Black	981	(917; 1046)	−82	0.108
	Boys	1044	(975; 1112)			White	1064	(996; 1132)		
BMI-FA z-score	Girls	1004	(949; 1059)	44	0.316	Black	939	(879; 999)	−173	<0.001
	Boys	1048	(981; 1115)			White	1112	(1050; 1174)		
FFM (kg)	Girls	1031	(977; 1086)	−25	0.582	Black	977	(919; 1036)	−91	0.039
	Boys	1006	(939; 1074)			White	1069	(1008; 1129)		
FFMI (kg/m^2^)	Girls	1034	(975; 1094)	−31	0.549	Black	961	(901; 1020)	−128	0.004
	Boys	1004	(930; 1078)			White	1089	(1027; 1151)		
FM (kg)	Girls	993	(938; 1049)	70	0.124	Black	934	(874; 994)	−182	<0.001
	Boys	1063	(995; 1131)			White	1116	(1053; 1179)		
FMI (kg/m^2^)	Girls	991	(935;1048)	74	0.107	Black	931	(870;992)	−189	<0.001
	Boys	1066	(997;1135)			White	1120	(1056;1184)		
Average Steps/day	Girls	1000	(942;1057)	55	0.244	Black	936	(870;1001)	−180	<0.001
	Boys	1054	(983;1125)			White	1116	(1046;1185)		

WFA Weight-for-age; HFA Height-for-age; BMI-FA Body mass index-for-age; FFM Fat-free mass; FFMI Fat-free mass index; FM Fat mass; FMI Fat mass index. ^a^ Girls *n* = 56; boys *n* = 38; black *n* = 49; white *n* = 45. ^b^ Adjusted mean: Predictive margins of the general linear model for ANOVA with the factors sex and population group and their interaction. ^c^ 95% Confidence interval around the mean. ^d^ Two-way ANOVA. ^e^ Multivariate regression analysis.

## Data Availability

The data presented in this study are available on request from the corresponding author.

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
