# Peer review of "Resting Energy Expenditure and Related Factors in 6- to 9-Year-Old Southern African Children of Diverse Population Groups"

_nutrients, 2021, doi:10.3390/nu13061983_

Round 1
Reviewer 1 Report
This is a well written and interesting paper. The authors explore the possibility that resting energy expenditure may explain differences in overweight/obesity seen between black South African and white South African population groups (in children). They also explore potential factors that may explain resting energy expenditure including body composition and physical activity.
The results are interesting and show that there is a difference in resting energy expenditure between black and white South African children. The authors suggest it may explain differences in BMI/overweight/obesity between population groups.
The methodology is appropriate. Although body composition measurements by DXA and physical activity using accelerometers would be ideal, the authors acknowledge this limitation. BIA and pedometers are acceptable in such a field setting. The lack of variation in socio-economic class within the sample may be an issue.
The analyses appear sound. However, there is one particular analysis I would recommend. I think it would be worthwhile to examine for an effect of the interaction between sex and population group on resting energy expenditure. I realise the study may not be powered to detect the effect of an interaction but it would be very interesting.
The second point I would make is that the authors need to discuss possible reasons for the differences in resting energy expenditure between the population groups, given that they do not appear to be explained by differences in body composition or physical activity.
Overall, this is an interesting paper and the suggestions are intended to enhance the paper's value.
Reviewer 2 Report
The manuscript written by Adeline Pretorius and colleagues addresses an interesting topic. The study aims to determine, by sex and population group, the resting energy expenditure of black and white African children. There are some comments that the authors should address.
Abstract
Please revise, at the end of the abstract, the sent4ence: “When adjusted for FFM and HFA z-score, the differences in 25 REE declined but remained significant at 91kcal/day (P=0.039) and 82kcal/day (P=0.108), 26 respectively.” Is P=0.108 still statistically significant?
Background
Introduction section is well written. Nevertheless, in the objective, it is stated that the authors aimed to compare black African and white South African children. I am wondering if black children are only from South Africa. If not, why do authors chose black African and white South African children. Are they comparable?
Methods
Methodology section is well written. I have the same question about the origin of the children African Vs South African. On the other hand, after reading this section I understood that South Africa did not refer to the country, rather than to the southern part of the African continent. I find this confusing in the form it is written.
Results
Sociodemographic data could be included in the article. If not, a sentence similar to “data not shown” should be added to the section.
Discussion
Discussion section is well written.
